# Prognostic Value of the Selected Polymorphisms in the *CD36* Gene in the Domain-Encoding Lipid-Binding Region at a 10-Year Follow-Up for Early-Onset CAD Patients

**DOI:** 10.3390/biomedicines11051332

**Published:** 2023-04-30

**Authors:** Michał Bartoszewicz, Monika Rać

**Affiliations:** 1Psychosocial and Medical Rehabilitation Centre, West Pomeranian Centre Oncology, 71-730 Szczecin, Poland; 2Department of Biochemistry and Medical Chemistry, Pomeranian Medical University, 70-204 Szczecin, Poland

**Keywords:** polymorphism *CD36*, CAD patients, ten-year follow-up

## Abstract

The polymorphism of the *CD36* gene may have a decisive impact on the formation and progression of atherosclerotic changes. The aim of the study was to confirm the prognostic values of the previously studied polymorphisms in the *CD36* gene within a 10-year follow-up period. This is the first published report confirming the long-term observation of patients with CAD. The study group covered 100 early-onset CAD patients. It included 26 women not older than 55 years and 74 men not older than 50 years, tested in a ten-year study as a long-term follow-up after the first cardiovascular episode. There are no notable differences between the *CD36* variants and the number of fatalities during observation, fatalities due to cardiological reasons, cases of myocardial infarction within a ten-year observation period, hospitalizations for cardiovascular issues, all cardiovascular occurrences, and the number of months lived. We have shown that the *CD36* variants analyzed in this study do not appear to be related to the risk of early CAD occurrence in the Caucasian population in long-term observation.

## 1. Introduction

Cardiometabolic diseases are more likely to develop when medical conditions like dysglycemia, dyslipidemia, obesity, and elevated blood pressure are present. Impaired cardiometabolic risk factors can pose a risk for years before any clinical symptoms arise, which can make effective management by cardiologists more challenging. The pathogenesis of atherosclerosis is multifactorial and diverse, entailing immunological and inflammatory processes and mechanical damage to the vascular endothelium. As it is commonly known, genetic factors are involved in developing cardiometabolic risk independently of external factors. Monocytes and macrophages, whose surfaces carry, among other things, scavenger receptors such as CD36 and others responsible for binding oxidized LDL (oxLDL) particles and their accumulation by macrophages, influence atherosclerosis pathomechanism.

A defining feature of dyslipidemia at the biochemical level is the elevation of oxidative stress, which results in the buildup of oxidized lipids in the walls of blood vessels, specifically in the form of oxidized low-density lipoproteins (oxLDL), and in the plasma. There is a link between dyslipidemia and elevated platelet activation. In individuals with acute coronary syndrome, the platelets circulate with oxLDL bound to their surface, indicating a functional connection to the disease. The oxLDL levels in the circulation have been found to predict forthcoming atherothrombotic events, including CAD, stroke, acute coronary syndrome, dyslipidemia, and metabolic syndrome. Studies have revealed that oxLDL affects the early stages of atherosclerosis, such as endothelial dysfunction due to endothelial cell network formation and dysregulation of endothelial nitric oxide production. As a result, the modified cholesterol species accumulation derived from oxLDL in macrophages and leading to the formation of foam cells in the intima plays a crucial role in driving pro-inflammatory processes in the vessel wall [1]. In the animal study, the downregulation of scavenger receptors serving as ligands for oxLDL greatly reduced the burden of atherosclerosis [2]. By contrast, the impairment of the phagocytic capacity of animal macrophages caused by reducing CD36 levels through matrix metalloproteinase (MMP-9)-dependent signaling leads to higher mortality in post-myocardial infarction [3].

The CD36 receptor, also named platelet glycoprotein IIIb or IV, is one of the most crucial membrane proteins present on the surface of a wide variety of cell types, including macrophages, adipocytes, hepatocytes, skeletal and cardiac myocytes, microvascular endothelial cells, platelets, kidney, breast, microvascular endothelial cells, and epithelial cells in the gut [4]. A study [5] on nearly 55,000 genes that make up signaling modules found that the *CD36* gene and three other genes were identified as key molecules targeted for treating patients with CAD. Two signaling pathways with the most relevance to inflammation and lipid formation were established. All these factors relate to the cellular oxidation process, suggesting that it has to identify therapeutic targets in a way related to pathways, including the CD36 receptor, which is a scavenger receptor involved in lipid metabolism. The transmembrane receptor named CD36 is mainly expressed on the platelet surface and is a receptor for oxLDL. It has been observed that there is an essential link between oxidative lipid stress, CD36, and platelet hyperactivity, which leads to a prothrombotic phenotype with the deletion of *CD36* [6]. The model of CD36 signaling in platelets and macrophages is presented in Figure 1. OxLDL binds to CD36 in a region between amino acids 155–183, which leads to the activation of a tyrosine kinase signaling pathway initiated by the activation of the Src family kinases (SKF) that link CD36 to multiple downstream signaling pathways. There is an association between SFK signaling, increased ROS generation, and platelet procoagulant activity [7,8,9]. OxLDL remarkably increases interleukin gene transcription, protein synthesis, and secretion in human macrophages. That activation is dependent on the oxLDL-induced generation of ROS too. The oxLDL molecules induced by ROS production and that mediate interleukins maturation depend mainly on the receptor of CD36 but not on other scavenger receptors of type SR-A. CD36 is involved in atherosclerosis progression. It promotes oxLDL-mediated inflammation and foam cell formation [10]. The functional response to oxLDL molecules can be influenced by the surface expression levels of the CD36 receptor. It may vary by as much as 50 times [11]. Moreover, it was demonstrated [12] that the receptor PCSK9 directly enhanced platelet activation and in vivo thrombosis by binding to the CD36 receptor in platelets. This is the induction of activation in downstream signaling pathways, independent of the LDL-related pathway. It further aggravates microvascular obstruction and infarct expansion after myocardial infarction (MI). Thus, oxLDL ligation to CD36 promotes platelet activation, generates procoagulant platelets, blunts inhibitory signaling, and contributes to a dyslipidemia-induced procoagulant state.

Understanding the expression profiles and functionality of the receptors in disease cohorts and a long-term follow-up of CAD patients may be critical to linking the CD36 receptor functions to cardiovascular diseases. Given this context, our earlier studies published in Gene [13] showed no differences in the frequencies of *CD36* variants between neonates and early-onset CAD patients. Only a borderline difference was found in the frequencies of rs141680676, i.e., the A591T genotypes of *CD36*. Between the neonates’ group and early CAD patients, there was also a borderline-important difference in the prevalence of haplotype CGCGCGT with both alleles: rs3173798 (IVS3-6C) and rs14161680676 (591T). Thus, we have concluded that the variants of the *CD36* gene analyzed in the Caucasian population, i.e., rs143150225, rs3211892, rs38897347, rs141680676, rs5956, rs3173798, and C311T, do not appear to be associated with the risk of CAD occurrence. The *CD36* polymorphism may, however, affect modifiable cardiovascular risk factors. Our other studies [14] on the *CD36* region encoding the receptor domain participating in oxLDL and long-chain fatty acid (LCFA) binding and metabolism show that in patients with early-onset CAD, the rs3173798 in the *CD36* gene is linked to the risk of myocardial infarction at an earlier age. Other polymorphisms, such as rs5956 and rs3173798, are attributable to a higher prevalence of left ventricular diastolic dysfunction or higher left ventricular mass (rs141680676) [15]. The rs5956 in the *CD36* gene has been associated with lower density and thickness of the atheromatous plaque, which suggests its protective effects against the development of atherosclerosis; however, it also implies a risk of plaque instability. The rs141680676 polymorphism, in turn, correlates with low ABI values that are a risk factor for cardiovascular events [16]. It is suggested that the polymorphism of the *CD36* gene may play a decisive role in the formation and progression of atherosclerotic changes. The analyses demonstrate that CD36 plays a role in developing vascular comorbidities. The findings suggest that addressing CD36 could be a preventive approach to managing vascular risk factors. Therefore, the current study aimed to analyze the predictive values of the previously studied polymorphisms in the *CD36* gene within a 10-year follow-up period. This is the first published report with such long-term observation of patients with CAD.

## 2. Materials and Methods

This study is a continuation of the previous research conducted in 2007–2010 and supported by the Ministry of Science and Higher Education (grant number: 2P05D 002 30). The research aimed to determine the effects of selected *CD36* polymorphisms on CAD risk factors and clinical parameters. The study group comprised 100 patients (74 men and 26 women) hospitalized in the Department of Cardiology, Marie Curie Hospital, Szczecin, Poland, diagnosed with coronary artery disease, where CAD diagnosis criteria as previously described [15] were either past myocardial infarction, history of cardiac revascularization procedure, or angiographically documented coronary artery stenosis reaching ≥40% of the left main coronary artery, or ≥50% of a major epicardial artery, or ≥70% of a branch. The study included patients in clinically stable condition, receiving optimal pharmacological treatment, who had not undergone acute coronary syndrome or revascularization procedures within the previous month. The study excluded patients with hemodynamically significant congenital or acquired valvular heart disease, symptomatic heart failure (NYHA class > 1), renal failure (creatinine > 3 mg/dL), type 1 diabetes mellitus, thyroid dysfunction (current hypo- or hyperthyroidism), or malignancy. The exclusion criteria included all coexisting hemodynamically significant congenital or acquired heart diseases and all co-morbidities that would significantly affect the patient’s clinical picture. The first study comprised men up to 50 years old and women up to 55 years old. The patients’ and controls’ (DNA 300 newborns) results of genetic material analysis for *CD36* polymorphisms performed during the previous project form the basis for the planned studies. For our unscreened (random) control group [17,18], we selected newborns from the same region as our study patients. This group is as unbiased as possible and provides an optimal estimate for the frequency of alleles in the general population. Although the age and gender of the control group do not match those of the study group, we had previously found no association between these variables and *CD36* polymorphisms in CAD patients [13]. Genomic DNA was isolated, and amplicons of exons 4, 5, and 6, including fragments of introns, were studied using the denaturing high-performance liquid chromatography (DHPLC) technique. Finally, heterozygous control probes from DNA fragments harboring sequence variants (when available) were used to confirm the efficiency of detecting genetic alterations. The PCR products with different alterations detected by DHPLC were bidirectionally sequenced using the Applied Biosystems Dye-terminator Cycle Sequencing Ready Reaction kit according to the manufacturer’s protocols previously described [19]. A 373A DNA fragment analyzer (Applied Biosystem, Foster City, CA) was used for semi-automated sequence analysis. To assess the conformity of the genotype distribution to the Hardy-Weinberg law, the exact test was applied [20].

To conduct a long-term follow-up, the present study tested the same group of patients who had experienced a cardiovascular episode ten years prior. The study collected data on the living patients’ health status, including medical documentation of hospitalizations for cardiovascular conditions, any cardiovascular events over the ten years, or any deaths caused by cardiovascular reasons. Our research has been conducted following the principles outlined in the Helsinki Declaration and has obtained the required approval from our institutional ethics committee (approval number: BN-001/162/04). We also obtained informed consent from each participant in the study. The next stage is to compare the association of *CD36* polymorphisms with the patient’s present health status or death for cardiovascular reasons. Genotype frequencies with qualitative variables were analyzed using a comprehensive Fisher test. One of the authors subjected the results to a statistical analysis using the Statistica 6.0 PL software. The *p* < 0.05 was determined to be statistically significant.

## 3. Results

In the previous study [19], we detected by DHPLC six single nucleotide substitutions. We focused on four of them for further analysis in our ten-year follow-up with CAD cases. We found the presence of IVS3-6 T/C (rs3173798) and IVS4-10 G/A (rs3211892) polymorphisms in intron fragments adjacent to the tested exons. We found that rs3173798 was the most common sequence alteration in all study groups. In early-onset CAD (9.5%) and newborn controls (9.6%), the IVS3-6C allele frequency was similar. In comparison, according to the NCBI dbSNP database, it was insignificantly elevated in the no-CAD adult group (13.75%) as compared to the earlier described Caucasian populations (6.2–10%). The IVS4-10A allele frequency in early-onset CAD (3.5%) and newborn controls (3.9%) was slightly higher. However, it was similar in the no-CAD adult group (1.25%) to that described in Caucasians (0.8–2.7%), according to the dbSNP. The overall frequency of the IVS3-6C allele (9.98%) was similar, while the IVS4-10A allele (3.59%) was insignificantly higher than that previously described in Caucasian populations, according to the NCBI dbSNP database. The synonymous transition G573A in exon 6 (Pro191Pro, rs5956) was the most common sequence alteration in the exons tested. Another synonymous alteration was A591T (Thr197Thr, rs141680676). In early-onset CAD patients, the 573A allele frequency was found to be 3.0%, 2.6% in newborn controls, and 6.25% in non-CAD adults. It is worth noting that the dbSNP database reports a range of 2.7–4.5% for the Caucasian populations. The dbSNP database does not quote the 591T allele’s frequency for the Caucasian population, while in the cohort population, it was 0.2%. The overall frequency of the 573A allele (3.59%) was similar, while the frequency of 591A (0.90%) was insignificantly higher than that quoted in the NCBI dbSNP database. There were rare nonsynonymous alterations in exon 6: C311T (Thr104Ile), G550A (Asp184Asn and rs138897347), and C572T (Pro191Leu and rs143150225), as each of them was detected in only one subject. Up until now, there has not been any description of the alteration of C311T in the literature. The 550A and 572T alleles’ overall frequencies (0.11%) were similar to those described earlier in the cohort population (0.1–0.2%), according to the NCBI dbSNP database. There was no deviation from the Hardy-Weinberg equilibrium (*p* > 0.05) noted for the *CD36* genotypes in the study group and in the two control groups concerning all sequence changes (*p* = 0.5942 for rs3173798 in the CAD group and *p* = 1 for the rest). The statistical power of the study was sufficient enough to detect true differences in rare allele frequencies between the CAD and newborn groups with 80% probability, which were equal to: −6% or +8% for rs3173798; −4% or +6% for rs3211892; and +3% to +5% for other polymorphisms. As per the HapMap database, rs10499859 (A/G substitution in 5′UTR) is in weak linkage disequilibrium with rs5956 (D′ = 1, r^2^ = 0.053, LOD = 2.37), rs3173798 (D′ = 1, r^2^ = 0.039, LOD = 1.59), and rs3211892 (D′ = 1, r^2^ = 0.017, LOD = 0.82). For this reason, our study focuses on the most common polymorphisms found in the region of the *CD36* gene that encodes the oxLDL binding domain. In this study on early-onset CAD, we have not observed any statistically significant differences between the variables in newborns and *CD36* polymorphisms in patients with CAD. Therefore, the occurrence of the tested polymorphisms in this group appears to be comparable to that in the general population. The distribution of the frequency of *CD36* sequence alterations in newborn controls compared to the general population is shown in Table 1.

By conducting this research, we may gain further insight into the CD36 receptor’s role in early atherosclerotic lesion formation in vessels leading to CAD. A follow-up after ten years is a pioneering study. The study focuses on the analysis of the selected *CD36* gene alterations (Table 2): single nucleotide substitutions in intron 3 rs3173798 (IVS3-6C), intron 4 rs3211892 (IVS4-10 G/A), exon 6 rs5956 (synonymous transition G573A and Pro191Pro), and rs141680676 (transversion A591T and Thr197Thr). Regarding the *CD36* genotypes observed in the study group, there was no deviation from Hardy-Weinberg equilibrium (*p* > 0.05).

The study group’s clinical characteristics and biochemical parameters are presented in Table 3.

In the present study, as seen in Table 4, there are no notable differences between the *CD36* variants and the number of fatalities during observation, fatalities due to cardiological reasons, instances of myocardial infarction within a ten-year observation period, hospitalizations for cardiovascular issues, all cardiovascular occurrences, and the number of months lived. None of the previous studies by other authors have shown the association of any variants of *CD36* with the examined parameters over such a long period of observation.

## 4. Discussion

Thus far, there is no explanation of the functional effects of the detected polymorphisms. The rs3173798 polymorphism is located at a conserved splice site [21,22]. In the African American population, the impact of the rs3173798 SNP on *CD36* expression is inversely related to its influence on serum HDL, which tends to increase *CD36* expression [23]. To date, no other publications have looked into the relationship between the genotypes studied and CD36 function from a biological point of view. The authors reported only one CD36 deficiency and cardiomyopathy case [24], observed for two years. The abovementioned medical condition involves reversible left ventricular (LV) dysfunction that is not limited to the distribution of the epicardial coronary artery. In fact, the performed coronary angiography showed no significant coronary stenosis in the analyzed case. Nonetheless, LV angiography demonstrated basal wall hyperkinesis and dyskinesia of the distal ½ of the LV, including the apex. Furthermore, flow cytometry revealed no *CD36* expression on monocytes or lymphocytes, indicating CD36 deficiency. The patient was then monitored for two years, and there were no cardiovascular events during that time. There was also an association between the C/T substitution in exon 4 (rs75326924) in the *CD36* gene and a significant decrease in LCFA uptake in hypertrophic cardiomyopathy, myocardial infarction, and dilated cardiomyopathy in the Japanese population [25,26], yet rs75326924 was not reported in the Caucasian population. Some single-nucleotide variants located in the non-coding region of the *CD36* gene could indirectly alter the protein’s expression and function, but there is no clear direction. Parra et al. [27] showed that the rs1194182G > C variant in the *CD36* gene provides a protective effect with a 1.7-fold lower susceptibility to developing acute coronary syndrome (*p* = 0.03); however, this association is masked by diabetes and dyslipidemia. On the other hand, CD36 deficiency, diagnosed as having type 1, based on a lack of uptake of 15-(p-iodophenyl)-3-R,S-methyl-pentadecanoic acid in myocardia, might cause extensive atherosclerosis leading to acute myocardial infarction [28]. In patients with coronary artery disease (CAD), an increased plasma LCFA level was found to be associated with certain *CD36* polymorphisms in the 5′UTR and 3′UTR flanking regions (rs1984112, rs1761667, rs1527483, rs3840546, rs1049673), as reported by Ma et al. [29]. In their research, there was a suggestion that these alterations may be linked to a potential risk of CAD. There is an association between the *CD36* promoter rs2151916 polymorphism and LDL levels that was demonstrated by Goyenechea et al. in the St. Thomas UK Adult Twin Registry cohort [30]. Moreover, other authors showed [31] that in Caucasians, rs3173798 alteration was not significantly linked to myocardial infarction. Another study [32] has been conducted on the association between single-nucleotide polymorphisms (SNPs) in the *CD36* gene, including rs1761667 (A/G substitution), and cardiovascular diseases, type 2 diabetes, metabolic syndrome, consumption of total fat, fat taste perception, or obesity. These studies have been performed on various ethnic populations around the world. However, ambivalent results regarding the association between the genotype distribution of the rs1761667 polymorphism and cardiovascular risk factors were obtained.

According to the findings of a genome-wide association study (GWAS) investigating the presence of CAD [33], no correlations were found with the *CD36* gene. Still, the GWAS of the four large cohorts (19,602 white people, including 1544 stroke cases) showed instead that *CD36* rs3211928 was significantly associated with stroke [34]. Long-term follow-up is needed to definitively resolve the controversy over the association of the *CD36* gene defect with cardiovascular disease. Based on our thorough research, we can confidently state that this study reports the necessary cornerstone information pertaining to this particular observation.

## 5. Conclusions

In conclusion, we have proven that the *CD36* variants analyzed in this study do not seem to be related to the risk of early CAD occurrence in the Caucasian population in long-term observation. Nonetheless, the current research has some drawbacks, including the absence of a control group without early-onset CAD, a homogenous study population, and a limited number of cases. The lack of associations does not resolve doubts about the relationship between *CD36* gene polymorphism and parameters of atherosclerosis progress manifested in subsequent cardiovascular episodes. Additional investigation is required to determine the link between the *CD36* gene and CAD risk. It is crucial to verify these findings in a more extensive sample of patients.

## Figures and Tables

**Figure 1 biomedicines-11-01332-f001:**
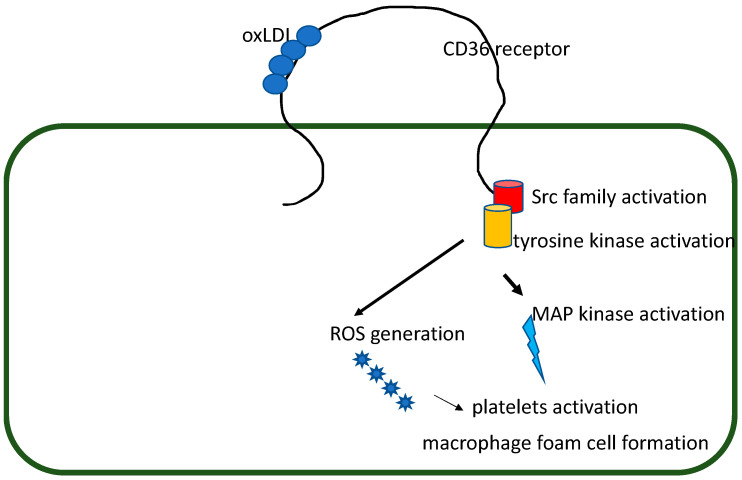
Model of CD36 signaling in platelets and macrophages.

**Table 1 biomedicines-11-01332-t001:** *CD36* sequence alterations frequency in newborn controls compared to the general population.

*CD36* Exon/Intron (Position) rs Number DNA Sequence/Protein Alteration	Genotype/Minor Allele Frequency		
	Allele	Newborns	dbSNP Base
intron 3 (80285850) rs3173798 IVS3-6 T/C	TT	249 (81.3%)	
	TC	55 (18.0%)	
	CC	2 (0.70%)	
	T	553 (90.4%)	93.8–90%
	C	59 (9.60%)	6.2–10%
intron 4 (80290369) rs3211892 IVS4-10 G/A	GG	282 (92.2%)	
	GA	24 (7.80%)	
	G	588 (96.1%)	92.3–99.2%
	A	24 (3.90%)	0.8–2.7%
exon 5 (80290408) no rs number C311T/Thr104Ile	CC	306 (100%)	
	CT	0 (0%)	
	C	612 (100%)	
	T	0 (0%)	lack of data
exon 6 (80292426) rs138897347 G550A/Asp184Asn	GG	305 (99.7%)	
	GA	1 (0.30%)	
	G	611 (99.8%)	99.8–99.9%
	A	1 (0.20%)	0.1–0.2%
exon 6 (80292448) rs143150225 C572T/Pro191Leu	CC	305 (99.7%)	
	CT	1 (0.30%)	
	C	1 (0.20%)	99.8–99.9%
	T	1 (0.20%)	0.1–0.2%
exon 6 (80292449) rs5956 G573A/Pro191Pro	GG	290 (94.8%)	
	GA	16 (5.20%)	
	G	596 (97.4%)	95.5–97.3%
	A	16 (2.60%)	2.7–4.5%
exon 6 (80292467) rs141680676 A591T/Thr197Thr	AA	303 (99%)	
	AT	3 (1.00%)	
	A	609 (99.5%)	99.8%
	T	3 (0.50%)	0.2%

**Table 2 biomedicines-11-01332-t002:** Distribution of the frequency of *CD36* sequence alternation in patients with CAD.

Position in *CD36* Exon/Intron Number of rs DNA Sequence Protein Alteration	Genotype	
	Allele	
intron 3 (80285850) rs3173798 IVS3-6 T/C	TT	81 (81.0%)
	TC	19 (19.0%)
	CC	0 (0%)
	T	181 (90.5%)
	C	19 (9.50%)
intron 4 (80290369) rs3211892 IVS4-10 G/A	GG	93 (93.0%)
	GA	7 (7.00%)
	G	193 (96.5%)
	A	7 (3.50%)
exon 6 (80292449) rs5956 G573A/Pro191Pro	GG	94 (94%)
	GA	6 (6.00%)
	G	194 (97.0%)
	A	6 (3.00%)
exon 6 (80292467) rs141680676 A591T/Thr197Thr	AA	96 (96%)
	AT	4 (4.00%)
	A	195 (98.0%)
	T	4 (2.00%)

**Table 3 biomedicines-11-01332-t003:** Clinical characteristics of the study group and its biochemical parameters.

Parameters	Value
% of males	74%
Age of CAD patients (years)	49.9 ± 5.91
Systolic BP (mmHg)	127 ± 14.0
Diastolic BP (mmHg)	77.0 ± 9.0
MAP (mmHg)	93.8 ± 9.4
WHR	0.96 ± 0.09
BMI (kg/m^2^)	28.1 ± 4.0
Waist (cm)	98.3 ± 12.5
Hip (cm)	103 ± 9
Hypertension	66%
Age of hypertension diagnosis (years)	42.6 ± 8.6
MI	70%
Age of the first MI (years)	44.0 ± 5.6
Current smoking	15%
Past smoking	89%
Years of smoking	18.9 ± 9.8
PTCA	71%
CABG	37%
Statins	96%
Beta-blockers	88%
ACEI	80%
ARB	17%
Calcium channel blockers	18%
Diuretics	31%
hsCRP (mg/L)	1.20 ± 0.27
glucose (mg/dL)	101 ± 2.49
CHc (mg/dL)	163 ± 4.06
HDL (mg/dL)	47.0 ± 1.16
LDL (mg/dL)	93.0 ± 3.64
TG (mg/dL)	128 ± 5.74
LP(a) (mg/dL)	20.3 ± 4.96
ApoA1 (mg/dL)	146 ± 3.85
ApoB (mg/dL)	74.0 ± 2.25
ApoB/ApoA1	0.52 ± 0.02
VEGF (pg/mL)	236 ± 17.2
IL-6 (pg/mL)	1.69 ± 2.77
Platelets (G/L)	216 ± 4.58
CD36 µg/mL	15.78 ± 12.9

Data are given as percentage or mean ± SD. MAP—mean arterial pressure; WHR—waist-to-hip ratio; BMI—body mass index; BP—blood pressure; MI—myocardial infarction; PTCA—percutaneous transluminal coronary angioplasty; CABG—coronary artery bypass grafting; ACEI—angiotensin 1 converting enzyme inhibitors; ARB—angiotensin 2 receptor blockers; hsCRP—high-sensitivity C-reactive protein; CHc—total cholesterol; HDL—high-density lipoprotein cholesterol; LDL—low-density lipoprotein cholesterol; TG—triglycerides; ApoA1, ApoB, and Lp(a)—apolipoproteins; IL-6—interleukin 6; VEGF—vascular endothelial growth factor; CD36—soluble CD36 protein in plasma.

**Table 4 biomedicines-11-01332-t004:** The analysis of the clinical data for CAD cases stratified by *CD36* genotype.

Parameter	IVS3-6 T/C			IVS4-10 G/A			Exon 6 G573A			Exon 6 A591T		
	TT n = 81	TC n = 19		GG n = 93	GA n = 7		GG n = 94	GA n = 6		AA n = 96	AT n = 4	
			*p*			*p*			*p*			*p*
Death during observation	8	1	1.00	8	1	0.51	8	1	0.46	9	0	1.00
Death for cardiological reasons	6	1	1.00	7	0	1.00	6	1	0.37	7	0	1.00
Myocardial infarctions during the 10-year observation	11	2	1.00	13	0	0.59	12	1	0.59	13	0	1.00
All cardiovascular events	23	3	0.38	25	1	0.67	25	1	1.00	26	0	0.56

## Data Availability

All data is available from the correspondent author, due to patient privacy policies.

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
