# Peer review of "Prognostic Value of the Selected Polymorphisms in the *CD36* Gene in the Domain-Encoding Lipid-Binding Region at a 10-Year Follow-Up for Early-Onset CAD Patients"

_biomedicines, 2023, doi:10.3390/biomedicines11051332_

Round 1

Reviewer 1 Report

The manuscript titled 'Prognostic value of the selected CD36 gene polymorphisms in the region encoding lipid-binding domain in 10 years follow-up in early-onset CAD patients' is well-designed and of importance because of the long-term follow-up of CAD patients and, therefore, of great clinical interest. However, additional clarifications are needed in the manuscript to improve the understanding of the data presented.

The authors write that the aim of the manuscript is to confirm the prognostic values of previously studied polymorphisms in the CD36 gene within a 10-year follow-up period. However, the authors of previous studies have not confirmed the association of CD36 polymorphisms with early-onset disease CAD. Therefore, the phrase 'confirm' may be too strong to use in a written research aim. 'Investigate' or 'analyze predictive values' instead of 'confirm' would be more appropriate.

In the Methods chapter, it is not necessary to describe the control group of 300 newborns whose DNA was analyzed because these data are not related to the patients and have no bearing on the prognostic significance of CD36 polymorphisms in patients with CAD.

The results written in lines 162-183 need to be clarified. Indeed, the study includes 100 CAD patients, and the results of newborn control subjects are also presented, which have nothing to do with the aim of this study. Moreover, because of the lack of clarity, it might be better to present them in a table and explain their significance for the interpretation of the results of the 10-year follow-up of CAD patients.

Some minor corrections are also needed:

Line 20 - is 'tirne' a typo? Should it be time?

Line 53 - no space between CD and 36. Correct to CD36

Lines 80-81 - in vivo should be italicized.

Line 107 - is 'in tum' a typo? Should it be written 'in turn'?

Figure 1 - the typo 'MAP kinase actication' in 'MAP kinase activation' should be corrected.

Line 200 - add 'exon 6' before SNP rs5956 to clarify.

Line 256 - the typo 'weas' should be corrected to 'was'.

Author Response

We are grateful to the reviewers for the valuable comments regarding the manuscript ID: biomedicines-2222895 entitled "Prognostic value of the selected CD36 gene polymorphisms in the region encoding lipid-binding domain in 10 years follow-up in early-onset CAD patients." submitted by Bartoszewicz and Rac to the special issue “Cellular Mechanisms of Cardiovascular Disease 2.0” of “Biomedicines” journal. The point-by-point responses to the reviewers` comments are presented below. The changes introduced to the manuscript are marked red.

Reviewer 1

The authors write that the aim of the manuscript is to confirm the prognostic values of previously studied polymorphisms in the CD36 gene within a 10-year follow-up period. However, the authors of previous studies have not confirmed the association of CD36 polymorphisms with early-onset disease CAD. Therefore, the phrase 'confirm' may be too strong to use in a written research aim. 'Investigate' or 'analyze predictive values' instead of 'confirm' would be more appropriate.

According to the reviewer`s suggestion we modified phrase in the aim of study.

In the Methods chapter, it is not necessary to describe the control group of 300 newborns whose DNA was analyzed because these data are not related to the patients and have no bearing on the prognostic significance of CD36 polymorphisms in patients with CAD.

The controls are usually subjects matched with age and gender to the study group, yet without the studied disease. The problem is that such controls frequently do not represent “general population”, because they are usually not randomly selected from the population but recruited from patients diagnosed or treated due to other symptoms or diseases, not associated with the studied disease. Lack of CAD symptoms would not prove the absence of the disease anyway, while presence of non-specific chest pains might lead to exclusion of some subjects without CAD and to further deterioration of the group representativity. The reference test - coronary angiography - could not be performed in controls for ethical reasons. It is recently proposed to circumvent such problems by choosing unscreened (random) control group (Ann Hum Genet. 2005, 69: 566-576 or Hum Hered. 2006, 61:22-26). The group composed of consecutive newborns from the same region where the studied patients live, seems as much unbiased as possible, and optimal to estimate frequency of an allele in the general population. So, we added such genetic control to the recent study. The arguments for choosing newborns as the control for the study were added and presented in the “Material and methods” section with new corresponding reference: We chose newborn population as unscreened (random) genetic control group [17-18]. The group composed of consecutive newborns from the same region where the studied patients live, seems as much unbiased as possible, and optimal to estimate frequency of alleles in the general population. Its age and gender are not matched to the study group, but we have not found any association between these variables and CD36 polymorphisms in CAD patients in our previous study [13].

The results written in lines 162-183 need to be clarified. Indeed, the study includes 100 CAD patients, and the results of newborn control subjects are also presented, which have nothing to do with the aim of this study. Moreover, because of the lack of clarity, it might be better to present them in a table and explain their significance for the interpretation of the results of the 10-year follow-up of CAD patients.

This part of section was added at the request of the editor. However, according to the reviewer`s suggestion we added new table in the “Results” section of study, so we changed the numbers of tables too. For the clarity we also added the sentences: In previous study [19] we detected by DHPLC 6 single nucleotide substitutions. Four of them we chose for further analysis in ten years follow-up with CAD cases. And also: We have not found statistical differences between these variables in newborn and CD36 polymorphisms in CAD patients. Which means that the frequency of the polymorphisms studied in the group in this study - early onset CAD is similar to the general population.

Some minor corrections are also needed:

Line 20 - is 'tirne' a typo? Should it be time?

Should be “months number of life”, without time. It was corrected.

Line 53 - no space between CD and 36. Correct to CD36

It was corrected.

Lines 80-81 - in vivo should be italicized.

It was corrected.

Line 107 - is 'in tum' a typo? Should it be written 'in turn'?

It was a typo, corrected.

Figure 1 - the typo 'MAP kinase actication' in 'MAP kinase activation' should be corrected.

It was corrected.

Line 200 - add 'exon 6' before SNP rs5956 to clarify.

It was added.

Line 256 - the typo 'weas' should be corrected to 'was'.

It was corrected.

I hope that the revised manuscript will prove suitable for publication in the journal “Biomedicines”.

With kind regards,

Monika Rac

Reviewer 2 Report

This study by Michał Bartoszewicz and Monika Rać aimed to investigate the prognostic values of 12 polymorphisms in the CD36 gene in the formation and progression of atherosclerotic changes in early-onset coronary artery disease (CAD) patients over a 10-year follow-up period. The study included 100 CAD patients, consisting of 26 women not older than 55 years and 74 men not older than 50 years. The results showed that there were no significant differences between the CD36 variants and the number of deaths, myocardial infarctions, cardiovascular hospitalizations, all cardiovascular events, and months of life during the 10-year follow-up period. The study concludes that the CD36 variants analyzed do not appear to be related to the risk of early CAD occurrence in the Caucasian population in the long-term.

General comments:

The study involved a long-term follow-up of patients with CAD, which is a strength of the study. However, the study is limited by its small sample size and lack of diversity in the study population. Additionally, the study did not investigate the functional significance of the CD36 variants, and the conclusions are based on their association with cardiovascular events. Overall, while the study provides valuable insights into the association between CD36 variants and cardiovascular events in early-onset CAD patients, further research is needed to confirm the findings and explore the functional implications of these variants (which is also mentioned by the authors in the discussion). Overall the study is well-written, although the results would perhaps benefit from from being breaken up into smaller sections.

Author Response

Response to the reviewer’s comments

         We are grateful to the reviewers for the valuable comments regarding the manuscript ID: biomedicines-2222895 entitled "Prognostic value of the selected CD36 gene polymorphisms in the region encoding lipid-binding domain in 10 years follow-up in early-onset CAD patients." submitted by Bartoszewicz and Rac to the special issue “Cellular Mechanisms of Cardiovascular Disease 2.0” of “Biomedicines” journal. The point-by-point responses to the reviewers` comments are presented below. The changes introduced to the manuscript are marked red.

Reviewer 2

General comments:

The study involved a long-term follow-up of patients with CAD, which is a strength of the study. However, the study is limited by its small sample size and lack of diversity in the study population. Additionally, the study did not investigate the functional significance of the CD36 variants, and the conclusions are based on their association with cardiovascular events. Overall, while the study provides valuable insights into the association between CD36 variants and cardiovascular events in early-onset CAD patients, further research is needed to confirm the findings and explore the functional implications of these variants (which is also mentioned by the authors in the discussion). Overall the study is well-written, although the results would perhaps benefit from from being breaken up into smaller sections.

Unfortunately, we have not studied the functional significance of the CD36 variants. According to the reviewer`s suggestion we added the sentences to the end of “Disscution” section with new corresponding reference:

The rs3173798 polymorphism is located at a conserved splice site [19-20] and the effect of the rs3173798 SNP on CD36 expression is inversely related to the impact on serum HDL, tended to increase CD36 expression in African American population [23]. The bio-logical basis of the relationship between CD36 function and the studied genotypes re-mains unknown and has yet to be reported in other publications.

We also added one more limitation “lack of diversity in the study population” to the “Conclusion” section.

I hope that the revised manuscript will prove suitable for publication in the journal “Biomedicines”.

With kind regards,

Monika Rac

Round 2

Reviewer 1 Report

The authors addressed all raised issues, and the manuscript is acceptable for publication.